# Exploring the relationships between eco-anxiety, eco-guilt, eco-grief, and pro-environmental behavior in the Dutch and German population: A cross-sectional study

Michele Petkovski[1]*, Johannes Steinrücke[2], Alejandro Dominguez-Rodriguez[1]

1 Section Psychology, Health and Technology, University of Twente, Enschede, the Netherlands,
2 Section Cognition, Data and Education, University of Twente, Enschede, the Netherlands

* c.m.petkovski@utwente.nl

## Abstract

While experiencing individual eco-emotions, such as eco-anxiety, eco-guilt, and eco-grief, has been linked to pro-environmental behavior, no prior studies have jointly investigated these variables. Research in Dutch and German populations is particularly scarce despite being at relatively high risk for experiencing the effects of climate change, such as floods. This study examined the relationship between eco-anxiety and pro-environmental behavior, with eco-guilt and eco-grief as mediators, and age, gender, and proximity to water as moderating variables. Cross-sectional data ($n = 311$) were collected using an online survey. Data analyses revealed significant positive correlations between all three eco-emotions and pro-environmental behavior. A positive relationship between eco-anxiety and pro-environmental behavior was found, which was mediated by eco-guilt, but not eco-grief. With a negative indirect effect of eco-guilt on pro-environmental behavior, eco-guilt was a non-linear, suppressor-like variable. Only age moderated the pathway from eco-anxiety to eco-guilt; no moderation effects were found for gender or proximity to water. This research provides preliminary evidence of the complex relationships between eco-anxiety, eco-guilt, eco-grief, and pro-environmental behavior in a Dutch and German population. The findings highlight the importance of developing educational programs to inform individuals about eco-emotions and potential coping strategies, while promoting pro-environmental behavior. Future studies with larger, more diverse samples are recommended to replicate the results and explore which groups of individuals may be more vulnerable to experiencing higher levels of eco-emotions. Further, intensive longitudinal research designs combined with (generalized) causal mediation analyses could be applied to unravel the temporal interplay of eco-emotions and PEB.

**Data availability statement:** An anonymized version of the data that support the findings of this study is available at the DANS Data Station via https://doi.org/10.17026/SS/3V2FMA.

**Funding:** The author(s) received no specific funding for this work.

**Competing interests:** The authors have declared that no competing interests exist.

## Introduction

Climate change is recognized as one of the largest health threats worldwide [1,2]. While gradual changes in the environment naturally occur, it has been established that human activity has significantly accelerated the process [3,4]. The IPCC [3] reports that global temperatures have heightened more rapidly from 1970 to 2020 than in any other 50-year period over the past 2,000 years. Primarily the emission of greenhouse gasses has been identified as a major cause of enhanced global warming, which is induced by, for instance, industrial activities, mass vehicle use, agriculture, and deforestation [5,6].

Some of the most widely known consequences of global warming on the physical environment include higher frequency of heat waves, more and longer droughts, severe precipitation, rising sea levels, and increased incidence of natural disasters such as hurricanes [5,7,8]. The current projections for the future show that extreme weather events will continue to advance [3,7,9]. A significant impact of this extreme weather, which has affected millions of people across Europe, is flooding [10]. Floods are considered the most prevalent natural disaster, and their frequency and intensity are anticipated to heighten in the following years [8]. People who live close to rivers, in coastal areas, and in lowland areas are especially at risk, as such locations are prone to flooding after severe precipitation [11,12]. Countries such as the Netherlands and Germany are particularly vulnerable, as much of the Netherlands is directly located along the sea, and several major rivers run through both countries. For instance, the flooding disaster in Western Europe in July 2021 resulted in damage exceeding 383 million Euros in the Netherlands [13] and 189 casualties and damage of over 33 billion Euros in Germany [14]. Thus, while both countries have a variety of flood prevention strategies, such as dikes along rivers, rapid environmental changes demand continuous adaptation in terms of water management [15]. Furthermore, in addition to alterations in the physical environment, climate change also has implications for mental well-being. Frequently reported direct consequences include acute stress responses and post-traumatic stress disorders, depressive symptoms, anxiety, and grief due to changes in one's direct environment [16,17].

### Eco-anxiety, eco-guilt, and eco-grief

Important experiences that have emerged regarding mental health issues related to climate change include eco-anxiety, eco-guilt, and eco-grief, collectively referred to as eco-emotions [18]. Firstly, the most frequently studied construct is eco-anxiety. There are currently various definitions and methods in which the phenomenon is measured, indicating a need for more conceptual clarity and consensus on its key aspects [19,20]. However, eco-anxiety is often described as a persistent fear or worry about the destruction of the environment [21]. Uncertainty, unpredictability, and uncontrollability of climate change significantly contribute to this phenomenon [21]. Prior studies suggest that eco-anxiety is a consequence of becoming aware of climate change, rather than experiencing its direct effects [22]. Nevertheless, there is a lack of consensus in the available literature on whether eco-anxiety should be considered a natural or pathological occurrence [18].

Secondly, eco-guilt can be characterized as guilt that arises when performing activities that are polluting the environment or when not taking action for the environment despite perceived personal or societal pressure to do so [23]. Nielsen et al. [24] found that perceived required knowledge and opportunities to act more sustainably are necessary factors for experiencing eco-guilt and that it only arises when the person wants to conform to their or society's norms. Furthermore, research indicates that levels of eco-guilt are higher when individuals consider their ecological footprint to be greater than average [18].

Lastly, eco-grief refers to feelings of grief caused by experienced or anticipated ecological loss resulting from climate change [25]. It can be evoked by actual losses of the physical environment, ecological knowledge, or expected future environmental changes [25] and may arise from specific smaller losses or the overall, global climate disruption [26]. Therefore, eco-grief is considered a natural response, and particularly individuals who feel closely connected to nature may experience more eco-grief [25].

While these three emotions appear to be distinct constructs, studies have repeatedly found that they positively correlate to each other (e.g., [18,27,28]).

## Eco-emotions and pro-environmental behavior

While there is no overall consensus yet on whether eco-anxiety is a natural or pathological phenomenon [18], Verplanken and Roy [29] suggest that eco-anxiety is an adaptive response to climate change. They argue worrying is a natural response to potential future threats, and may prompt emotion regulation behaviors, such as pro-environmental behavior (PEB) [29]. As positive correlations between the three eco-emotions have previously been found, eco-guilt and eco-grief could operate in similar ways.

Accordingly, eco-anxiety, eco-guilt, and eco-grief have all individually been related to PEB. It can be defined as behaviors that are designed to decrease or avoid negative human consequences on the environment, such as recycling [30]. Ágoston et al. [18] propose that certain levels of eco-emotions may lead to environmental action as individuals likely to experience eco-emotions are also more oriented towards the climate. Studies show that people who experience more eco-anxiety exhibit more PEB [18,31–34]. Interestingly, Whitmarsh et al. [35] found that eco-anxiety only predicts some PEBs, such as stimulating others to conserve energy; behaviors like reducing meat consumption and purchasing items with less packaging, among others, were not associated with eco-anxiety. Furthermore, experiencing higher eco-guilt has been repeatedly linked to increased PEB [18,36–38], as well as intentions to act more environmentally friendly [23]. However, some studies have found that PEB only increases when feelings of eco-guilt subconsciously serve as a reminder of how individuals should have behaved, but not in other circumstances [24]. Additionally, while positive associations between eco-grief and PEB have been found [18,37], research on the relationship between these two variables is limited, highlighting a need for further investigation.

## Influences of age, gender, and proximity to water

Multiple factors that can influence eco-emotions and PEB have been identified. For instance, various studies have demonstrated that particularly younger individuals and women are more vulnerable to developing higher levels of eco-anxiety [18,28,31,32,39]. Younger people may be more psychologically affected because they are more likely to experience the effects of climate change during their lifetime, while women reportedly worry more frequently about the implications of climate change [19]. Similarly, higher levels of eco-guilt are found among younger individuals compared to older individuals, and women tend to experience more eco-guilt and eco-grief than men [18,38].

Moreover, PEB and age have been positively associated in various studies [40–42]. There also appear to be generational differences in the types of PEBs; for instance, age groups below 40 more frequently report using green transportation compared to those above 40, while age groups above 50 report more boycotting against products created by environmental offenders compared to younger age groups [30]. Additionally, prior research indicates gender differences in relation to PEB, however, findings across studies are inconsistent. For example, Patel et al. [41] found that males exhibit

more PEB than females, whereas others indicate that women engage or intend to engage in PEB more than men [43,44]. Conversely, another study showed no gender differences regarding PEB [45]. These contrasting findings highlight that the relationship between gender and PEB is complex and possibly multi-layered.

Due to the increasing risk of floods (e.g., [5]), another variable of interest is how close people live to larger water bodies, such as a sea, lake, or river. For instance, a positive correlation has been established between proximity to the seacoast and a river, and eco-anxiety [46]. Furthermore, in an experimental study by Fox et al. [47], participants virtually navigated their avatars along a polluted river. They were randomly assigned to one of two conditions: either they were informed that they were near (in the same city) or far away (2,000 miles) from that river. Their results indicated that individuals in the near condition felt psychologically closer to the river, which led to increased risk perception and, consequently, more PEBs in the days following the experiment compared to people in the far away condition [47]. These findings suggest that closeness to water may be a relevant factor; however, other research about the effects of proximity to water on eco-emotions and PEB appears to be currently nonexistent.

## The current study

While prior research has shown associations between eco-anxiety, eco-guilt, eco-grief, and PEB, most studies have only included two of these variables (e.g., eco-anxiety and PEB [35]). Others have investigated all four but only examined simple relations between them, for instance, between all eco-emotions separately and PEB [18], or they correlated eco-anxiety, eco-guilt, and eco-grief to each other [18,28]. Currently, there seem to be no studies that jointly analyze the three eco-emotions and PEB to examine potential mediating relationships. Consequently, there is little understanding of how these emotions may collectively relate to PEB, and through which mechanisms such behavior may be motivated. We hypothesize that as eco-anxiety results from becoming aware of climate change [22], it could be considered an immediate and natural response. Persistent thoughts about the destruction of the environment [21] may then lead to increased awareness of one's (lack of) activities supporting the environment (resulting in eco-guilt), and/or awareness of current or potential future losses in the physical environment (resulting in eco-grief). To then regulate such thoughts and emotions, people may engage in more PEB. Including all four variables in one model allowed us to explore these interrelationships between eco-anxiety, eco-guilt, eco-grief, and PEB.

Additionally, research that includes age, gender, and proximity to water as moderating variables on the relationships between eco-emotions and PEB is still limited. Findings from prior studies together indicate that for younger people and women, associations between the eco-emotions and PEB may be stronger than for older people and men. Further, particularly the mixed evidence on the gender influences on PEB, and the lack of studies on closeness to water suggest that more research is required in this area. Lastly, as previously described, various countries, such as the Netherlands and Germany, are at an increased risk of flooding due to their coastal areas, lowlands, and rivers [11,12]. However, only a few studies on eco-emotions and PEB have been conducted among German populations (e.g., [28,34]) and, to the best knowledge of the authors, there is currently no scientific literature available regarding Dutch populations. This is important to note, as the experience of eco-emotions and relationships between them could be different in the Netherlands and Germany compared to other countries.

Considering these literature gaps, this study aimed to explore the associations between eco-anxiety, eco-guilt, eco-grief, and PEB while also considering the potential roles of age, gender, and proximity to water. The research question this paper will be centered around is: "To what extent are eco-anxiety and pro-environmental behavior related, considering eco-guilt and eco-grief as mediators and age, gender, and proximity to water as moderators?" Moderating effects will also be examined on the potential mediating pathways. The hypotheses that will be tested are: (1) eco-anxiety and pro-environmental behavior are positively related, (2) eco-guilt and eco-grief mediate the relationship between eco-anxiety and pro-environmental behavior, and (3) age, gender, and proximity to water moderate the mediated relationship between eco-anxiety and pro-environmental behavior. The hypotheses are visualized in Fig 1.

The potentially moderating effects of age, gender, and proximity to water (H3) will be tested on all pathways.

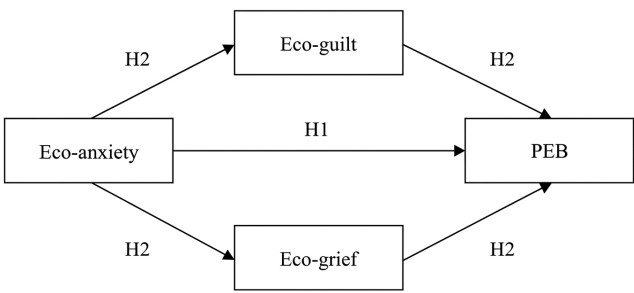

**Fig 1. Hypothesized Associations.**

## Methods

### Study design and participants

This cross-sectional study was executed through an online survey administered via Qualtrics. While cross-sectional data does not allow to infer causality [48], we used (moderated) mediation models to explore preliminary relationships between the eco-emotions and PEB. The outcomes are therefore presented as descriptive associations, as they may not accurately reflect causal relationships [49]. To enhance completeness and accuracy in reporting this paper, the Strengthening the Reporting of Observational Studies in Epidemiology (STROBE) guidelines were applied [50], see S1 Table.

Participants were recruited through convenience and snowball sampling. The questionnaire was shared with individuals in the research team's networks (e.g., directly or via social media platforms such as LinkedIn), who were also asked to distribute the survey link within their networks. Furthermore, the questionnaire was uploaded to SurveyCircle (www.surveycircle.com), a platform specifically designed to recruit participants for online research studies, and to Sona Systems at the University of Twente (www.sona-systems.com), an online test subject pool for research studies at universities. Inclusion criteria for this study were being 18 years or older, currently living in the Netherlands or Germany, and having a Dutch or German nationality. Individuals who were in treatment for a psychological disorder at the time of participation, had suicidal ideation within the last two years, or did not meet the inclusion criteria were ineligible for participation. Data were collected from 18 March to 29 June 2024. Fig 2 illustrates the sample filtering procedure from 427 recorded responses to 311 final participants. A post-hoc analysis using G*Power 3.1.9.6 [51] showed >0.99 statistical power at α = .05 ($f^2$ = 0.15) for the 311 remaining participants. This study was approved by the Ethics Committee of the Faculty of Behavioral, Management, and Social Sciences at the University of Twente (request number 240192).

The mean age of the participants was 29.3 (SD = 13.0), ranging from 18 to 75. Of the 311 participants, 207 were female, 102 were male, and 2 reported a third gender. Table 1 presents further demographic data on nationality, education level, and proximity to water.

### Materials

Participants completed the questionnaire on their own devices, such as a laptop or phone. Advertisements were designed to facilitate distribution on social media, which all contained a brief description of the research topic and the inclusion criteria. The survey was presented in Dutch or German; participants could select their preferred language. Demographic questions about age, gender, education level, and nationality were asked, as well as a binary question about whether participants live near bodies of water (e.g., sea, lake, river). Four psychological scales were included to measure the eco-emotions and PEB. The scales were independently forward-translated into Dutch and German by two German researchers using the online translation tool DeepL and subsequently revised by one Dutch and one other German native speaker. These revised versions were implemented in the study, but have not been validated (see Limitations). Since this

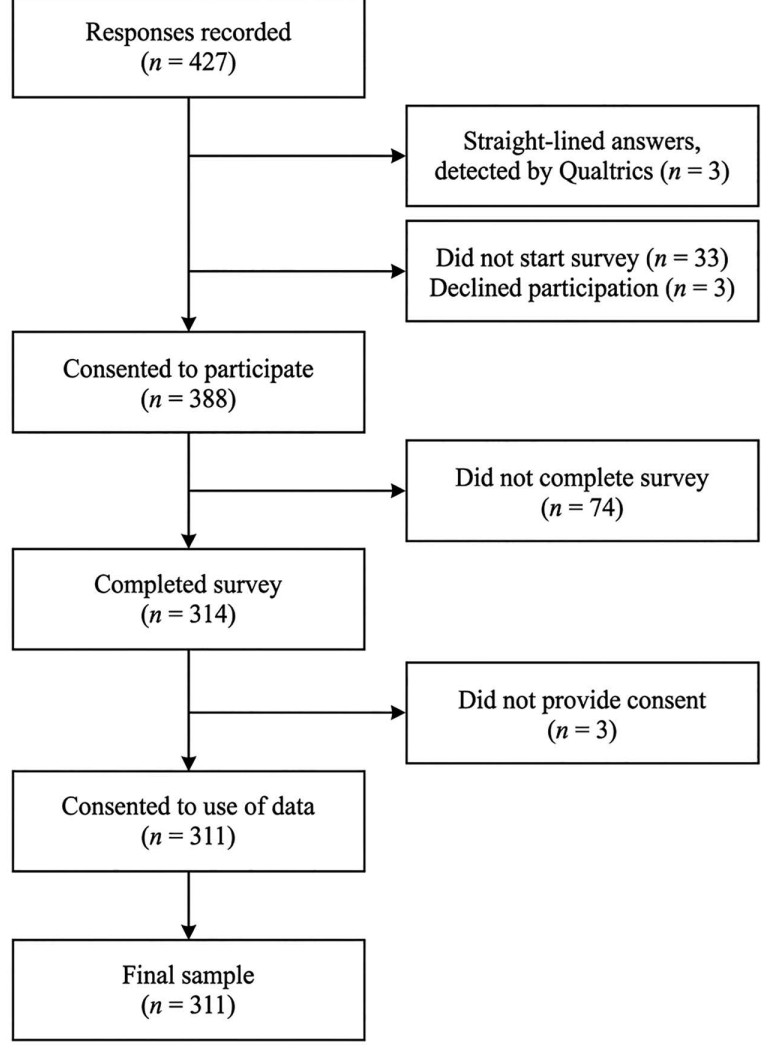

**Fig 2. Sample Filtering Flowchart.**

study was embedded in a larger project, additional items and psychological scales were incorporated. However, these were not relevant to this paper, thus, they are not reported here.

### Eco-anxiety questionnaire (EAQ-22)

The EAQ-22 was used to assess eco-anxiety. It consists of 22 items and has two subscales, distinguishing between habitual ecological worry and negative consequences of eco-anxiety. The questionnaire includes statements such as "I have unusual tension in my muscles since I've become more aware of climate change." Items were scored on a 4-point Likert scale ranging from 1 = "Strongly disagree" to 4 = "Strongly agree". Sum scores were obtained (ranging between 22 and 88), with higher scores indicating higher levels of eco-anxiety. While Ágoston et al. [18] only provide the internal consistency for the subscales, the current sample demonstrated excellent internal consistency for the overall scale ($\alpha = .93$).

**Table 1. Demographic Information.**

| Variable | n | % |
|---|---|---|
| **Nationality** | | |
| Dutch | 174 | 55.9 |
| German | 137 | 44.1 |
| **Education level** | | |
| High school | 89 | 28.6 |
| Secondary vocational or technical school | 19 | 6.1 |
| University of applied sciences | 42 | 13.5 |
| Undergraduate university degree (Bachelor's) | 108 | 34.8 |
| Graduate university degree (Master's) | 37 | 11.9 |
| PhD / doctoral degree | 6 | 1.9 |
| Other | 10 | 3.2 |
| **Living near water** (yes / no) | 199 / 112 | 64.0 / 36.0 |

*n* = 311.

### Eco-Guilt Questionnaire (EGuiQ-11)

The EGuiQ-11 was used to measure eco-guilt. This instrument includes 11 items, for instance, "It makes me feel uneasy that I am part of a system that is amplifying climate change." All items were answered on a 4-point Likert scale ranging from 1 = "Strongly disagree" to 4 = "Strongly agree". Total scores were acquired by summing the points of all items (ranging between 11 and 44); higher scores suggest higher levels of eco-guilt. The scale has shown high internal consistency in this sample ($\alpha$ = .90), replicating the original results ($\alpha$ = .89; [18]).

### Ecological Grief Questionnaire (EGriQ-6)

Eco-grief was examined with the EGriQ-6. It contains six statements, such as "It makes me sad that I don't see many of the plants and animals I used to see often." These were answered on a 4-point Likert scale, ranging from 1 = "Strongly disagree" to 4 = "Strongly agree". Sum scores were calculated to assess the level of eco-grief (ranging between 6 and 24). The higher the participants scored, the higher their eco-grief. High internal consistency was found for the current sample ($\alpha$ = .81), exceeding the original findings ($\alpha$ = .77; [18]).

### Pro-Environmental Behavior Scale (PEBS)

The PEBS was utilized to evaluate pro-environmental behavior. It consists of 19 items, categorized into four sub-scales: conservation, environmental citizenship, food, and transportation. An example item is: "During the past year, have you increased the amount of organically grown fruits and vegetables you consume?" Items were answered on Likert scales ranging from two (e.g., "Yes" and "No") to five points (e.g., "Never" to "Always"), with scores ranging from 1 to 5, depending on the item. Sum scores were calculated (ranging between 19 and 95); higher total scores indicate greater PEB. The overall scale demonstrated good internal consistency in this sample ($\alpha$ = .73) and the original sample ($\alpha$ = .80; [52]). Furthermore, the original questionnaire has high test-retest reliability (*r* = .85) and construct validity [52].

### Procedure

After accessing the survey, participants first viewed a welcome page that included the purpose and duration of the study, the inclusion and exclusion criteria, information about voluntary participation, the right to withdraw, anonymity,

confidentiality, and contact details of the researchers. Individuals who provided written consent to participate were then asked demographic questions and answered the question about proximity to water. Subsequently, the psychological scales were presented in fixed order: (1) EAQ-22, (2) EGuiQ-11, (3) EGriQ-6, and (4) PEBS. Afterwards, participants were debriefed, and in case of discomfort resulting from the survey, they were offered support via national suicide prevention lines. Contact information of the research team was again provided, and participants were asked if they still consented to their data being used after answering the questions. Lastly, participants could provide their email addresses if they were interested in receiving a summary of the study outcomes. It took approximately 15–20 minutes to complete the survey.

### Statistical analysis

R version 4.4.0 was used for data analysis [53]. Data were cleaned by removing the scales and variables that were not relevant to this paper and filtering out participants who did not meet the criteria for participation, did not provide informed consent, had not completed the survey, or straight-lined their answers. Subsequently, character variables were recoded into numeric or factor variables, depending on the variable. Next, total scores were calculated for each psychological scale, and descriptive statistics were assessed. Additionally, correlations between the variables were obtained.

Before hypothesis testing, the assumptions of linear regression (linearity, homoscedasticity, and normality of residuals) were examined for all hypotheses, and none were violated. The assumption of independence is assumed to be met due to the cross-sectional nature of this study and was therefore not tested. Consequently, for H1, a linear model was fitted to estimate the association between eco-anxiety and PEB. Moreover, for H2 and H3, PROCESS Macro version 4.3.1 [54] was used. For H2, model 4 was applied to test for multiple mediation of eco-guilt and eco-grief on the relationship between eco-anxiety and PEB. For H3, models 7 and 15 were used to examine moderated mediation effects of age, gender, and proximity to water. Model 7 tests for moderation on the a-paths of the mediation, and model 15 tests the b-paths and c'-path. Each of these models was run three times, as there are three hypothesized moderators, and PROCESS only allows for one moderator to be tested per model. 95% confidence intervals were computed using the standardized estimates if available, elsewhere the unstandardized estimates where used. Effect sizes in PROCESS are labelled either as *Effect* or *Index* (i.e., the regression coefficient $b$ or $β$), and are reported as such. A significance level of $α = .05$ was chosen for all hypotheses. In S2 Appendix, a visualization of the pathways of the moderated mediation model can be found.

## Results

### Descriptive statistics

Descriptive statistics of eco-anxiety, eco-guilt, eco-grief, and PEB, as well as bivariate correlations including age, can be found in Table 2.

Means and standard deviations for eco-anxiety, eco-guilt, eco-grief, and PEB are presented per category of the demographic variables in Table 3. Moreover, associations between eco-anxiety, eco-guilt, eco-grief, and PEB, and demographic variables (gender, nationality, and proximity to water) are visualized in boxplots, see S2-S4 Fig.

### Inferential statistics

**Eco-anxiety and PEB.** For H1, linear regression analysis was performed to examine the relationship between eco-anxiety and PEB. The model was significant, $R^2 = .26$, $F(1, 309) = 106.3$, $p < .001$. The results indicate a positive association between eco-anxiety and PEB, $β = 0.506$, $b = 0.436$, $SE = 0.042$, $t(309) = 10.31$, $p < .001$, 95% CI [0.353, 0.519].

**Table 2. Means, Standard Deviations, and Bivariate Pearson Correlations.**

| Variable | M (SD) | 1 | 2 | 3 | 4 |
|---|---|---|---|---|---|
| 1. Eco-anxiety | 52.6 (12.1) | – | – | – | – |
| 2. Eco-guilt | 24.5 (7.4) | 0.72* | – | – | – |
| 3. Eco-grief | 14.5 (4.1) | 0.79* | 0.57* | – | – |
| 4. PEB | 63.7 (10.4) | 0.51* | 0.29* | 0.45* | – |
| 5. Age | 29.3 (13.0) | −0.03 | −0.26* | 0.05 | 0.11 |

* $p < .001$.

**Table 3. Means and SDs Per Demographic Variable.**

| Variable | Eco-Anxiety | Eco-Guilt | Eco-Grief | PEB |
|---|---|---|---|---|
|  | M (SD) | M (SD) | M (SD) | M (SD) |
| Gender |  |  |  |  |
| Female | 54.1 (11.3) | 25.9 (7.2) | 15.1 (3.9) | 64.7 (10.2) |
| Male | 49.3 (13.0) | 21.7 (7.0) | 13.2 (4.2) | 61.6 (10.7) |
| Non-binary/other | 66.0 (12.7) | 28.0 (2.8) | 20.0 (1.4) | 73.5 (5.0) |
| Nationality |  |  |  |  |
| Dutch | 49.2 (12.2) | 23.4 (7.2) | 13.6 (4.1) | 61.5 (10.3) |
| German | 56.8 (10.6) | 26.0 (7.4) | 15.7 (3.8) | 66.5 (9.9) |
| Living near water |  |  |  |  |
| Yes | 53.4 (12.0) | 25.2 (7.0) | 14.7 (4.1) | 64.8 (10.3) |
| No | 51.1 (12.2) | 23.3 (7.9) | 14.2 (4.1) | 61.8 (10.4) |

**Eco-guilt and eco-grief as mediators.** For H2, PROCESS Macro model 4 was used to investigate the mediating effects of eco-guilt and eco-grief on the relationship between eco-anxiety and PEB. It was found that eco-guilt was a mediator, while eco-grief was not. When including mediators, model fit increased from $R^2 = .26$ without mediators to $R^2 = .28$ with mediators. The standardized indirect effects of eco-anxiety on PEB are presented in Table 4. Furthermore, the output for each pathway of the mediation model is reported in Table 5. Variance Inflation Factors for the complete model equaled 2.09, 95% CI [1.79, 2.50] for eco-guilt, 3.71, 95% CI [3.10, 4.51] for eco-anxiety, and 2.61, 95% CI [2.21, 3.14] for eco-grief, and can therefore be considered non-problematic [55].

The mediation analysis additionally revealed that even when eco-guilt is included in the model, the c'-path (eco-anxiety to PEB) shows a significant positive relationship. However, the indirect effect of eco-anxiety through eco-guilt on PEB is

**Table 4. Standardized Indirect Effects of Eco-Anxiety on PEB.**

|  | Effect | SE | LLCI / ULCI |
|---|---|---|---|
| Total | −0.009 | 0.077 | −0.165 / 0.139 |
| Eco-guilt | −0.118 | 0.050 | −0.219 / −0.019 |
| Eco-grief | 0.109 | 0.061 | −0.012 / 0.229 |

Headings are labelled according to PROCESS Macro output. LLCI and ULCI are the lower and upper bounds of the 95% Confidence Intervals. Results based on bootstrapping.

**Table 5. Findings Mediation Model Per Pathway.**

| Path | β | b | t(df) | p | LLCI / ULCI |
|---|---|---|---|---|---|
| a1 (eco-guilt) | 0.722 | 0.441 | 18.36(309) | <.001** | 0.394 / 0.489 |
| b1 (eco-guilt) | −0.163 | −0.230 | −2.33(307) | .021* | −0.425 / −0.036 |
| a2 (eco-grief) | 0.785 | 0.267 | 22.30(309) | <.001** | 0.244 / 0.291 |
| b2 (eco-grief) | 0.139 | 0.352 | 1.77(307) | .078 | −0.039 / 0.743 |
| c' (eco-anxiety) | 0.515 | 0.444 | 5.50(307) | <.001** | 0.285 / 0.602 |

The pathways are visualized in S1 Fig.* $p < .05$, ** $p < .001$. LLCI and ULCI are the lower and upper bounds of the 95% Confidence Intervals.

significant and negative. This is notable since eco-anxiety and eco-guilt, as well as eco-guilt and PEB, were all positively correlated (see Table 2). Accordingly, the relationship between eco-anxiety, PEB, and eco-guilt was visualized in a scatter-plot, see Fig 3. Considering both the S-curve and the reversal of the sign, eco-guilt seems to be a non-linear, suppressor-like variable in the relationship between eco-anxiety and PEB, resulting in a localized suppression, rather than a full range suppression.

**Age, gender, and proximity to water as moderators**

For H3, PROCESS Macro models 7 and 15 were run to test for moderated mediation effects of age, gender, and proximity to water on the relationship between eco-anxiety and PEB. It was found that only age moderated the mediation of eco-guilt on the eco-anxiety to eco-guilt path (i.e., a1-path), $b = -0.004$, $SE = 0.002$, $t(307) = -2.01$, $p = .046$, 95% CI [−0.007, < −0.001]. Table 6 illustrates the standardized effects of the six moderated mediation models; the unstandardized statistics for each pathway are presented in Table 7.

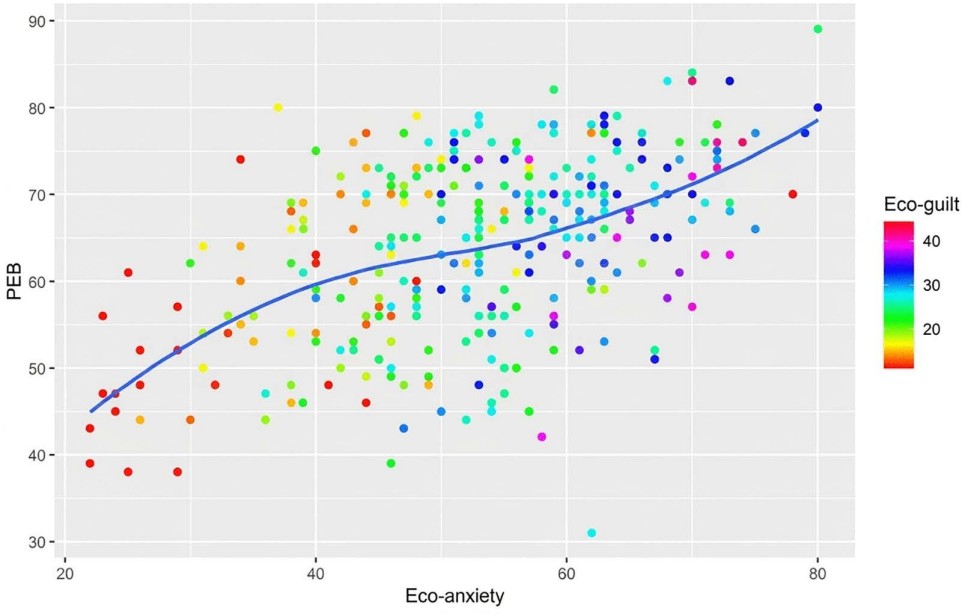

**Fig 3. Confounding Effect of Eco-Guilt on the Relationship Between Eco-Anxiety and PEB.**

**Table 6. Standardized Effects Moderated Mediation.**

| Variable | β | SE | LLCI | ULCI |
|---|---|---|---|---|
| | **a1-path (eco-guilt)** | | | |
| Age | 0.001 | 0.001 | 0.000 | 0.002 |
| Gender | −0.007 | 0.006 | −0.021 | 0.003 |
| Proximity to water | 0.015 | 0.014 | −0.008 | 0.049 |
| | **b1-path (eco-guilt)** | | | |
| Age | 0.006 | 0.004 | −0.001 | 0.014 |
| Gender | −0.046 | 0.045 | −0.135 | 0.045 |
| Proximity to water | 0.062 | 0.088 | −0.108 | 0.240 |
| | **a2-path (eco-grief)** | | | |
| Age | 0.000 | 0.000 | −0.001 | 0.001 |
| Gender | −0.002 | 0.005 | −0.013 | 0.007 |
| Proximity to water | −0.000 | 0.010 | −0.021 | 0.020 |
| | **b2-path (eco-grief)** | | | |
| Age | −0.006 | 0.004 | −0.014 | 0.002 |
| Gender | 0.075 | 0.059 | −0.042 | 0.192 |
| Proximity to water | 0.113 | 0.110 | −0.097 | 0.333 |

LLCI and ULCI are the lower and upper bounds of the 95% Confidence Intervals.

**Table 7. Findings Moderated Mediation Models Per Pathway.**

| Variable | b | t(df) | p | LLCI/ ULCI |
|---|---|---|---|---|
| | **a1-path (eco-guilt)** | | | |
| Age | −0.004 | −2.01(307) | .046* | −0.007/ −0.000 |
| Gender | 0.031 | 1.27(307) | .205 | −0.017/ 0.080 |
| Proximity to water | −0.063 | −1.27(307) | .205 | −0.162/ 0.035 |
| | **b1-path (eco-guilt)** | | | |
| Age | 0.014 | 1.79(303) | .075 | −0.001/ 0.030 |
| Gender | −0.105 | −0.95(303) | .344 | −0.322/ 0.113 |
| Proximity to water | 0.139 | 0.68(303) | .494 | −0.261/ 0.540 |
| | **a2-path (eco-grief)** | | | |
| Age | −0.000 | −0.08(307) | .938 | −0.002/ 0.002 |
| Gender | −0.006 | −0.49(307) | .622 | −0.031/ 0.018 |
| Proximity to water | −0.0010 | −0.03(307) | .975 | −0.050/ 0.048 |
| | **b2-path (eco-grief)** | | | |
| Age | −0.023 | −1.59(303) | .112 | −0.051/ 0.005 |
| Gender | 0.281 | 1.26(303) | .208 | −0.157/ 0.719 |
| Proximity to water | 0.422 | 1.01(303) | .314 | −0.402/ 1.245 |
| | **c'-path (eco-anxiety)** | | | |
| Age | −0.002 | −0.39(303) | .700 | −0.014/ 0.009 |
| Gender | −0.051 | −0.58(303) | .561 | −0.225/ 0.123 |
| Proximity to water | −0.274 | −1.59(303) | .112 | −0.611/ 0.064 |

\* $p < .05$. LLCI and ULCI are the lower and upper bounds of the 95% Confidence Intervals.

## Discussion

This study investigated the relationships between eco-anxiety, eco-guilt, eco-grief, and pro-environmental behavior (PEB) and the influence of age, gender, and proximity to water on these associations. The paper was centered around the research question "To what extent are eco-anxiety and pro-environmental behavior related, considering eco-guilt and eco-grief as mediators and age, gender, and proximity to water as moderators?". Findings indicate a significant association between eco-anxiety and PEB; eco-guilt was a mediator of this relationship, and age moderated one pathway of the moderated mediation model.

### Eco-anxiety and pro-environmental behavior

The first hypothesis states that eco-anxiety and PEB are positively related. Results indicated a significant positive association, supporting H1. This finding aligns with previous studies, which have similarly found that individuals with higher levels of eco-anxiety tend to exhibit more environmentally friendly behavior compared to those experiencing lower levels of eco-anxiety (e.g., [18,32,33]). The consistency between studies highlights that eco-anxiety and PEB are associated across different samples, suggesting it is a global phenomenon that requires further and more in-depth investigation.

### Mediation of eco-guilt and eco-grief

The second hypothesis, that eco-guilt and eco-grief mediate the relationship between eco-anxiety and PEB, has found only partial support: eco-guilt was a significant mediator, while eco-grief was not. No previous studies have examined the relationship between eco-anxiety and PEB with eco-guilt and eco-grief as mediators, however, current findings indicate that nearly all pathways of the mediation model were significant, except the path from eco-grief to PEB. This is in line with prior associations between eco-anxiety and eco-guilt [18,28], eco-guilt and PEB [36,38], and eco-anxiety and eco-grief [18,28]. Only the non-significant pathway between eco-grief and PEB contrasts with other research, which demonstrated a positive relationship between the two variables [18,37]. It is notable that the eco-grief to PEB path was non-significant, as these variables were moderately correlated in the initial bivariate correlation analysis. Consequently, eco-grief is not a mediator of the original relationship between eco-anxiety and PEB. This finding calls for an adjustment to the theoretical model proposed in Fig 1. As eco-grief was significantly related to eco-anxiety in the full mediation model, but not to PEB, this could imply that eco-grief may potentially explain the relationship between eco-anxiety and PEB, rather than lead to behavior in the way that eco-guilt might. While the methods in this study do not allow for testing causality, the findings of the mediation analyses could suggest a sequential model: experiencing eco-grief may make people aware of or reflect on climate change, which could trigger eco-anxiety, which in turn may affect eco-guilt, and may potentially result in exhibiting PEB. We recommend employing a research design capable of capturing changes in eco-emotions early on, such that the temporal order in which the eco-emotions change is identifiable. For instance, some form of intensive longitudinal research designs combined with (generalized) causal mediation analyses could be used.

Moreover, since eco-guilt was a significant mediator, eco-guilt may be a variable through which eco-anxiety influences PEB. While the relationship between eco-anxiety and PEB initially appeared linear, visual analyses revealed that including eco-guilt in the model leads to a more complex, non-linear type of relationship. Although at first this might seem like a suppression effect, which would imply that the relationship between eco-anxiety and PEB changes with changes in one's level of eco-guilt, this is not the case, as the overall magnitude of the direct relationship does not change. Further, the suppression seems to appear only locally around average eco-anxiety scores, meaning that more extreme levels of eco-anxiety are exempt from this effect.

Notably, as visualized in Fig 3, the overall relationship appears positive but also non-linear: The slope of the association differs across levels of eco-guilt. Firstly, for individuals with low levels of eco-guilt, the slope is increasing but flattening, suggesting that their experience of eco-anxiety may motivate them to act more environmentally friendly, or that their

eco-anxiety levels are low because they may already be performing PEBs. Secondly, for individuals with medium levels of eco-guilt, the slope is relatively stable, implying that their eco-anxiety levels may not motivate them to perform more PEB than they already are. Lastly, individuals with high levels of eco-guilt may perform more PEBs at an accelerating rate, possibly because their eco-anxiety levels become alarming to them, and potentially leading them to do as much as possible to counteract their anxiety and/or negative impact on the environment. Furthermore, when considering the eco-guilt scores in Fig 3, there appears to be weak negative relationships within the overall positive relationship. This may be explained by the reversal of the sign of the association between eco-guilt and PEB when comparing it with the linear model results from H1 and the c'-path coefficient in the mediation model. Overall, this non-linear, suppression-like effect indicates that the association between eco-anxiety and PEB could be more complex than previous research (e.g., [18,31,34]) suggests; there appears to be a curvilinear relationship that should be further examined. Also considering the findings concerning eco-grief, this too calls for additional analyses in the framework of (generalized) causal mediation analysis.

## Moderation effects of age, gender, and proximity to water

The third hypothesis states that age, gender, and proximity to water moderate the mediated relationship between eco-anxiety and PEB. As age moderated the path from eco-anxiety to eco-guilt, only limited support for H3 was found. Research has shown that younger people experience higher levels of eco-anxiety (e.g., [31,32]) and eco-guilt [18,38] in general, as they will likely personally experience the consequences of climate change in their lives [19]. This may explain the moderating effect of age, with younger people showing a stronger relationship between eco-anxiety and eco-guilt. However, the finding that age moderated only this pathway was unexpected, as previous studies have also found links between age and PEB [30,40,42]. This contrast is likely because (1) the current sample mainly consisted of younger adults, and/or (2) different age groups could exhibit different types of PEBs; the PEBS consists of four subscales, and response clusters on these subscales might have cancelled each other out when not differentiating between age groups on a subscale level.

Additionally, no gender differences were found. This contradicts most prior studies, as female gender has repeatedly been linked to experiencing higher levels of all three eco-emotions [18,28,39]. Furthermore, previous studies have shown inconsistent results regarding gender differences in PEBs, with some suggesting that males exhibit more PEB [41] and others demonstrating more PEB in females [43,44]. Nevertheless, the non-significant results in the current study align with the findings from Nuryadin et al. [45], highlighting that the role of gender in experiencing eco-emotions and exhibiting PEB is not yet well understood and could be influenced by other factors.

Lastly, no moderation effect was demonstrated for proximity to water. The lack of moderation likely occurred due to the phrasing of the question of whether participants live near water; participants' interpretation of the question may have influenced their answer and, consequently, the study's outcomes. While Parreira and Mouro [46] established an influence of closeness to water on eco-anxiety, no other studies have investigated this association or linked proximity to water to other eco-emotions. Moreover, an experimental study found preliminary evidence that participants who believed to be near a polluted river showed more PEB in the days after the experiment than those far away [47]. However, given the context and lack of further research, those outcomes may not be generalizable to real-life samples such as in the current study; thus, these results were not replicated.

## Limitations and future directions

Several limitations need to be considered when interpreting the outcomes. Firstly, the four psychological questionnaires used have not been validated in Dutch and German. Therefore, the results may not accurately measure the underlying constructs and generalizability is limited. However, the questionnaires were translated and evaluated by one Dutch and three German native speakers prior to this study; thus, validity is assumed to some extent. Furthermore, it is notable that a validated German version of the eco-emotion scales now exists [28]. As the article by Zeier and Wessa [28] was published after data collection for the current study started, it is recommended that future studies among German populations

include their version of the questionnaires. Additionally, the Dutch versions of the eco-emotion scales used in the current study, were recently validated in a follow-up study and are available in Dominguez-Rodriguez et al. [56], and can therefore also be applied in future research. Compared to the current study, only the answer options were slightly adapted to be more distinct. Validations of the PEBS in both languages are suggested.

Secondly, the phrasing of the question regarding participants' proximity to water, "Do you live near a body of water?" should be adjusted in future research on this variable. In the current study, the question did not specify an exact definition of closeness (e.g., distance in kilometers) or "a body of water." While some examples were provided on the latter, such as a river, no indication of the size of said river or other relevant characteristics was stated. This ambiguity led to the question being open to different interpretations and, consequently, unexplained variance in responses. Therefore, it is recommended to specify, for instance, that the water body must be large enough to cause nuisance if it were to flood. Additionally, reformulating the question to assess the distance to the nearest water body categorically (e.g., 0–5 km, 5–10 km) or continuously through exact distance rather than as a binary measure could provide more insight into how proximity to water affects the experience of eco-emotions and PEB. Investigating personal flood risk perception may also be relevant, as closer proximity to water may coincide with increased flooding experiences or risk perceptions and consequently greater intensity of eco-emotions or PEB.

Moreover, convenience and snowball sampling were used to recruit participants. While this method was chosen due to the limited time for data collection and thus increased efficiency, the resulting sample characteristics may have partly influenced the study outcomes. The mean age of the sample was 29.3 years old, with the majority being under the age of 35 ($n = 250$; 80.4%). The limited diversity across age groups within the sample could also explain the lack of significant results regarding age. Furthermore, participants' education level is considered high, as 65.3% have obtained a higher education degree. Consequently, the results should not be generalized beyond the groups represented in this study, which is notable as particularly little is known about the experience of eco-emotions in such groups. Future research should be conducted with a balanced age and educational sample, as differences have been identified in previous studies regarding pro-environmental behavior, eco-anxiety, eco-guilt, and eco-grief. For example, it has been found that younger people have higher eco-anxiety scores and are less likely to recycle, but more likely to use green transport, whereas older age groups are more likely to recycle and save resources, especially water [30]. Cooperating with research panels, for instance, through research organizations such as Statistics Netherlands (Centraal Bureau voor de Statistiek), could mitigate this limitation by ensuring balanced sample characteristics.

To the best knowledge of the authors, this was the first study to investigate more complex associations between eco-anxiety, eco-guilt, eco-grief, and PEB. Therefore, further studies with similar designs are needed to examine replicability across different samples and confirm the validity of the results. Additionally, as no inference can be made about the directionality of the associations due to the cross-sectional nature of this research [48], future studies with longitudinal designs are recommended to establish causality. Lastly, the found S-curve suggests that the relationship between eco-anxiety and PEB could be curvilinear and may resemble a cubic function. Therefore, using more complex modeling, such as cubic regression or (generalized) causal mediation analysis, rather than linear regression, may provide more accurate insight into this association. The confounding effect of eco-guilt also highlights a need to explore other factors related to the three eco-emotions, specifically to the relationship between eco-anxiety and PEB.

## Strengths

While previous studies have investigated simpler relationships between eco-anxiety, eco-guilt, eco-grief, and PEB, such as correlations, this was the first study to include all four in a single model (i.e., the mediation model). This analysis allowed for gaining a more comprehensive overview and preliminary evidence of how the variables are interrelated. Additionally, the confounding effect suggests that the association between eco-anxiety and PEB may be more complicated than initially expected; the results suggest that eco-guilt may play an important role and provide a new opportunity

for more in-depth research on these interactions. In addition to the mediation analysis, the potential influences of age, gender, and proximity to water were considered, with the latter being especially unique compared to prior research. Lastly, this was one of the first studies to investigate eco-emotions and PEB among a sample of individuals at an increased risk of flooding due to climate change. Therefore, the findings can contribute to understanding how people from countries such as the Netherlands and Germany may be psychologically affected by climate change.

### Practical applications

The outcomes of this study could be applied in multiple ways. Firstly, an overall positive relationship between eco-anxiety and PEB was found. Given that the direction of this association is currently unknown, it is unclear whether eco-anxiety leads to PEB or vice versa. Additionally, as eco-guilt was a mediator, its role may be relevant to consider. For instance, these findings could be used when developing educational programs that inform about the experience of eco-emotions and provide strategies for managing them while stimulating PEB. Particularly for individuals who experience low levels of eco-anxiety and eco-guilt, it may be relevant to focus on increasing PEB, as they could be less inclined or intrinsically motivated to perform PEBs. The mainly non-significant findings of the current research regarding age, gender, and proximity to water suggest that such educational programs should be broadly applicable to various populations.

However, it is relevant to investigate which groups of people may be more vulnerable to developing higher levels of eco-anxiety, eco-guilt, and eco-grief on a broader scale. Additionally, while PEB may serve as a coping mechanism for eco-emotions, further research is needed to explore other strategies. Therefore, studies with larger sample sizes and more diverse sociodemographic characteristics and geographic locations must be conducted to gain more insight into how climate change affects different populations. Findings from such studies could be applied to develop educational programs containing coping strategies tailored to specific at-risk groups.

### Conclusion

This study was the first to jointly investigate the relationships between eco-anxiety, eco-guilt, eco-grief, and pro-environmental behavior. The results suggested a positive association between eco-anxiety and PEB, which was mediated by eco-guilt but not eco-grief. Moreover, the moderating effects of age, gender, and proximity to water were examined; only age moderated the relationship between eco-anxiety and eco-guilt. Therefore, this research provides preliminary evidence of how the three eco-emotions and PEB may be interrelated. Nevertheless, future studies are needed to replicate the findings in larger and more diverse samples and thereby confirm the validity of the results.

### Supporting information

**S1 Table. STROBE statement for cross-sectional studies.**
(DOCX)

**S1 Fig. Pathways moderated mediation model.**
(DOCX)

**S2 Fig. Boxplots of gender and eco-anxiety, eco-guilt, eco-grief, and PEB.**
(DOCX)

**S3 Fig. Boxplots of nationality and eco-anxiety, eco-guilt, eco-grief, and PEB.**
(DOCX)

**S4 Fig. Boxplots of proximity to water and eco-anxiety, eco-guilt, eco-grief, and PEB.**
(DOCX)

## Author contributions

**Conceptualization:** Michele Petkovski, Johannes Steinrücke, Alejandro Dominguez-Rodriguez.

**Data curation:** Michele Petkovski, Johannes Steinrücke.

**Formal analysis:** Michele Petkovski, Johannes Steinrücke.

**Investigation:** Michele Petkovski.

**Methodology:** Michele Petkovski, Johannes Steinrücke, Alejandro Dominguez-Rodriguez.

**Project administration:** Alejandro Dominguez-Rodriguez.

**Supervision:** Johannes Steinrücke, Alejandro Dominguez-Rodriguez.

**Visualization:** Michele Petkovski, Johannes Steinrücke.

**Writing – original draft:** Michele Petkovski.

**Writing – review & editing:** Michele Petkovski, Johannes Steinrücke, Alejandro Dominguez-Rodriguez.

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
