## [Decision Letter · Decision Letter 0]

8 Mar 2026

PONE-D-25-59455Exploring the relationships between eco-anxiety, eco-guilt, eco-grief, and pro-environmental behavior in the Dutch and German population: A cross-sectional studyPLOS One

Dear Dr. Petkovski,

Thank you for submitting your manuscript to PLOS ONE. After careful consideration, we feel that it has merit but does not fully meet PLOS ONE’s publication criteria as it currently stands. Therefore, we invite you to submit a revised version of the manuscript that addresses the points raised during the review process.

**ACADEMIC EDITOR: Please revise**

We look forward to receiving your revised manuscript.

Kind regards,

Zhengmao Li

Academic Editor

PLOS One

Journal Requirements:

3. In the online submission form, you indicated that the data that support the findings of this study cannot be shared publicly because participants only provided consent for their data to be used for research purposes, and not for unrestricted public dissemination of individual-level data. However, a de-identified version of the dataset may be made available upon reasonable request. In this version, potentially identifying variables (e.g., exact ages or other personal characteristics) will be recoded into broader categories (e.g., age groups of 20–30 years) to further protect participant confidentiality. To obtain more information regarding the data collection and dataset, please contact Michele Petkovski via c.m.petkovski@utwente.nl. For further information regarding the ethical application, the BMS Ethics Committee of the University of Twente may be contacted via ethicscommittee-hss@utwente.nl.

Reviewers' comments:

Reviewer's Responses to Questions

**Comments to the Author**

1. Is the manuscript technically sound, and do the data support the conclusions?

Reviewer #1: Yes

Reviewer #2: Yes

2. Has the statistical analysis been performed appropriately and rigorously? 

Reviewer #1: Yes

Reviewer #2: Yes

3. Have the authors made all data underlying the findings in their manuscript fully available?

Reviewer #1: No

Reviewer #2: Yes

4. Is the manuscript presented in an intelligible fashion and written in standard English?

Reviewer #1: Yes

Reviewer #2: Yes

5. Review Comments to the Author

Reviewer #1: This study looks at how three eco emotions eco anxiety, eco guilt, and eco grief relate to pro environmental behavior in the Netherlands and Germany. It also examines whether living near water changes these relationships. The topic is important, and the paper is well written. The study is strong in looking at multiple emotions together and exploring a new idea about water proximity. A few edits and clarifications could make the paper even clearer and stronger.

Suggestions and Edits:

1. The abstract calls eco guilt a confounding variable, but it is better described as a suppressor variable if it changes the size or direction of the effect. This will make the meaning clearer.

2. Eco anxiety is described as natural or pathological. You could discuss whether the strong link with pro-environmental behavior suggests it is an adaptive response rather than a mental disorder.

3. The scales were translated into Dutch and German but are not formally validated. Describe how translation was done and mention the limitation.

4. Proximity to water was measured with a yes/no question, which may not capture real differences. Consider suggesting future research use exact distances or personal flood risk perception.

5. Eco anxiety and eco grief are highly correlated , which may indicate overlap. Report Variance Inflation Factor to show multicollinearity did not affect results. Also, discuss that grief may lead to reflection rather than active behavior, unlike guilt.

6. The sample is young and well educated, which may limit generalizability. Discuss how this could affect eco guilt or environmental actions.

Reviewer #2: This paper examines how eco-anxiety, eco-guilt, and eco-grief relate to pro-environmental behavior in a Dutch and German sample using mediation and moderated mediation analyses. Its main strength is that it puts these variables into one model and finds an interesting result around eco-guilt instead of only reporting simple correlations. The main issues are the cross-sectional design, the use of translated but not yet validated measures, and some interpretations that go beyond what the analyses can firmly support.

1. The paper asks a clear and meaningful question, and the overall model is worth studying. I like that Section 1.4 lays out the hypotheses clearly and that Fig. 1 makes the proposed relationships easy to follow. This gives the paper a solid structure from the start.

2. The strongest part of the manuscript is the eco-guilt result. In Section 3.2.1 and Tables 4 and 5, the indirect effect through eco-guilt is negative, even though the simple correlations are positive, which makes this finding genuinely interesting. At the same time, this part needs a clearer explanation because many readers may struggle to understand what this pattern really means.

3. I think the interpretation sometimes goes too far beyond the evidence. The study is cross-sectional, so it cannot support causal or sequential claims strongly, yet parts of Sections 4.2 and 4.6 move in that direction. The discussion of a possible S-shaped pattern and the idea of eco-guilt acting like a confounder are interesting, but they would need stronger statistical support.

4. The main limitations need to be taken more seriously in the framing of the conclusions. The sample was gathered through convenience and snowball sampling, the mean age was 29.3, most participants were female, and the translated Dutch and German scales were not yet validated. Because of this, the findings are promising, but they should be presented more cautiously and not generalized too broadly.

6. PLOS authors have the option to publish the peer review history of their article (what does this mean?). If published, this will include your full peer review and any attached files.

Reviewer #1: No

Reviewer #2: No

You may also use PLOS’s free figure tool, NAAS, to help you prepare publication quality figures: https://journals.plos.org/plosone/s/figures#loc-tools-for-figure-preparation

---

## [Author Response · Author response to Decision Letter 1]

21 Apr 2026

Reviewer #1: This study looks at how three eco emotions eco anxiety, eco guilt, and eco grief relate to pro environmental behavior in the Netherlands and Germany. It also examines whether living near water changes these relationships. The topic is important, and the paper is well written. The study is strong in looking at multiple emotions together and exploring a new idea about water proximity. A few edits and clarifications could make the paper even clearer and stronger.

Suggestions and Edits:

Comment: 1. The abstract calls eco guilt a confounding variable, but it is better described as a suppressor variable if it changes the size or direction of the effect. This will make the meaning clearer.

Response: Indeed, we could call eco-guilt a suppressor variable. However, that implies a linear relationship. Due to the S-curve visible in Fig. 3, we therefore chose to call it “non-linear suppressor-like”, highlighting the complexity of the relationship, while still acknowledging the “suppression” of the sign-flip. This change is applied in the abstract, results, and discussion.

Comment: 2. Eco anxiety is described as natural or pathological. You could discuss whether the strong link with pro-environmental behavior suggests it is an adaptive response rather than a mental disorder.

Response: Thank you for this comment. On page 5 in the section ‘Eco-emotions and pro-environmental behavior’, we have rephrased the first two sentences as follows:

“While there is no overall consensus yet on whether eco-anxiety is a natural or pathological phenomenon [18], Verplanken and Roy [29] suggest that eco-anxiety is an adaptive response to climate change. They argue worrying is a natural response to potential future threats, and may prompt emotion regulation behaviors, such as pro-environmental behavior (PEB) [29].”

Comment: 3. The scales were translated into Dutch and German but are not formally validated. Describe how translation was done and mention the limitation.

Response: We have added the translation procedure in more detail in the ‘Materials’ section in the methods (page 11), which now reads as follows:

“Four psychological scales were included to measure the eco-emotions and PEB. The scales were independently forward-translated into Dutch and German by two German researchers using the online translation tool DeepL and subsequently revised by one Dutch and one other German native speaker. These revised versions were implemented in the study, but have not been validated (see Limitations).”

The limitation is described in the discussion in the ‘Limitations and future directions’ section, please see the paragraph on page 25.

Comment: 4. Proximity to water was measured with a yes/no question, which may not capture real differences. Consider suggesting future research use exact distances or personal flood risk perception.

Response: Future directions regarding proximity to water were described in the ‘Limitations and future directions’ section (pages 25 and 26). However, we have added further details in response to this suggestion. The section relating to distance now reads as follows:

“Additionally, reformulating the question to assess the distance to the nearest water body categorically (e.g., 0-5 km, 5-10 km) or continuously through exact distance rather than as a binary measure could provide more insight into how proximity to water affects the experience of eco-emotions and PEB. Investigating personal flood risk perception may also be relevant, as closer proximity to water may coincide with increased flooding experiences or risk perceptions and consequently greater intensity of eco-emotions or PEB.”

Comment: 5. Eco anxiety and eco grief are highly correlated, which may indicate overlap. Report Variance Inflation Factor to show multicollinearity did not affect results. Also, discuss that grief may lead to reflection rather than active behavior, unlike guilt.

Response: Thanks for these suggestions. Regarding H2, on page 17, right above Table 4, we added that “Variance Inflation Factors for the complete model equaled 2.09, 95% CI [1.79, 2.50] for eco-guilt, 3.71, 95% CI [3.10, 4.51] for eco-anxiety, and 2.61, 95% CI [2.21, 3.14] for eco-grief, and can therefore be considered non-problematic.”

Additionally, we have added that eco-grief may lead to reflection and therefore potentially to eco-anxiety in the Discussion at section ‘Mediation of eco-guilt and eco-grief’ (page 22):

“As eco-grief was significantly related to eco-anxiety in the full mediation model, but not to PEB, this could imply that eco-grief may potentially explain the relationship between eco-anxiety and PEB, rather than lead to behavior in the way that eco-guilt might. While the methods in this study do not allow for testing causality, the findings of the mediation analyses could suggest a sequential model: experiencing eco-grief may make people aware of or reflect on climate change, which could trigger eco-anxiety, which in turn may affect eco-guilt, and may potentially result in exhibiting PEB. We recommend employing a research design capable of capturing changes in eco-emotions early on, such that the temporal order in which the eco-emotions change is identifiable. For instance, some form of intensive longitudinal research designs combined with (generalized) causal mediation analyses could be used.”

Comment: 6. The sample is young and well educated, which may limit generalizability. Discuss how this could affect eco guilt or environmental actions.

Response: We agree that this is a limitation, and it was mentioned in the previous version. However, in the revised version, we expanded this by including a recent study that presents differences in pro-environmental behavior and eco-emotions between younger and older people. A recommendation on how to overcome this limitation is also included in the manuscript, see page 26:

“Future research should be conducted with a balanced age and educational sample, as differences have been identified in previous studies regarding pro-environmental behavior, eco-anxiety, eco-guilt, and eco-grief. For example, it has been found that younger people have higher eco-anxiety scores and are less likely to recycle, but more likely to use green transport, whereas older age groups are more likely to recycle and save resources, especially water [30]. Cooperating with research panels, for instance, through research organizations such as Statistics Netherlands (Centraal Bureau voor de Statistiek), could mitigate this limitation by ensuring balanced sample characteristics.”

Reviewer #2: This paper examines how eco-anxiety, eco-guilt, and eco-grief relate to pro-environmental behavior in a Dutch and German sample using mediation and moderated mediation analyses. Its main strength is that it puts these variables into one model and finds an interesting result around eco-guilt instead of only reporting simple correlations. The main issues are the cross-sectional design, the use of translated but not yet validated measures, and some interpretations that go beyond what the analyses can firmly support.

Comment: 1. The paper asks a clear and meaningful question, and the overall model is worth studying. I like that Section 1.4 lays out the hypotheses clearly and that Fig. 1 makes the proposed relationships easy to follow. This gives the paper a solid structure from the start.

Response: Thank you for the comment, we appreciate this.

Comment: 2. The strongest part of the manuscript is the eco-guilt result. In Section 3.2.1 and Tables 4 and 5, the indirect effect through eco-guilt is negative, even though the simple correlations are positive, which makes this finding genuinely interesting. At the same time, this part needs a clearer explanation because many readers may struggle to understand what this pattern really means.

Response: Thank you for pointing out that this part needed more clarification. While referring to the abstract, reviewer 1 had a similar comment. There we clarified that we interpret eco-guilt as a non-linear, suppressor-like variable. We chose this phrasing, because “suppressor” would imply a linear relationship. Due to the S-curve visible in Fig. 3, we therefore chose to call it non-linear suppressor-like, highlighting the complexity of the relationship, while still acknowledging the “suppression” of the sign-flip. In accordance with your comment we clarified this again on page 18 (lines 361-364), and pages 22-23 (lines 422-429, 443-446).

Comment: 3. I think the interpretation sometimes goes too far beyond the evidence. The study is cross-sectional, so it cannot support causal or sequential claims strongly, yet parts of Sections 4.2 and 4.6 move in that direction. The discussion of a possible S-shaped pattern and the idea of eco-guilt acting like a confounder are interesting, but they would need stronger statistical support.

Response: Thanks for this suggestion. We indeed have limited statistical support due to the study design; we hypothesize that eco-guilt is a suppressor-like variable (previously referred to as confounder) because the data only allow hypothesizing rather than showing clear evidence. We have modified how we frame our findings/conclusions (e.g., changed ‘implies’ to ‘could imply’) throughout the Discussion and further addressed the limitations, especially that the study is cross-sectional and that we cannot support causal or sequential findings.

Comment: 4. The main limitations need to be taken more seriously in the framing of the conclusions. The sample was gathered through convenience and snowball sampling, the mean age was 29.3, most participants were female, and the translated Dutch and German scales were not yet validated. Because of this, the findings are promising, but they should be presented more cautiously and not generalized too broadly.

Response: We agree that in the previous version of the manuscript, especially in the Discussion, the limitations of our study were not adequately highlighted, given that our sample consists of mostly young, highly educated people. We reviewed the manuscript and made several changes to frame the findings differently. Additional literature was included to further explain why future research is needed, particularly with older generations, as there are considerable differences across age groups. Thanks for pointing this out.

---

## [Decision Letter · Decision Letter 1]

3 May 2026

Exploring the relationships between eco-anxiety, eco-guilt, eco-grief, and pro-environmental behavior in the Dutch and German population: A cross-sectional study

PONE-D-25-59455R1

Dear Dr. Petkovski,

We’re pleased to inform you that your manuscript has been judged scientifically suitable for publication and will be formally accepted for publication once it meets all outstanding technical requirements.

Kind regards,

Zhengmao Li

Academic Editor

PLOS One

Additional Editor Comments (optional):

Reviewers' comments:

Reviewer's Responses to Questions

**Comments to the Author**

1. If the authors have adequately addressed your comments raised in a previous round of review and you feel that this manuscript is now acceptable for publication, you may indicate that here to bypass the “Comments to the Author” section, enter your conflict of interest statement in the “Confidential to Editor” section, and submit your "Accept" recommendation.

Reviewer #1: All comments have been addressed

Reviewer #2: All comments have been addressed

2. Is the manuscript technically sound, and do the data support the conclusions?

Reviewer #1: Yes

Reviewer #2: Yes

3. Has the statistical analysis been performed appropriately and rigorously? 

Reviewer #1: Yes

Reviewer #2: Yes

4. Have the authors made all data underlying the findings in their manuscript fully available?

Reviewer #1: Yes

Reviewer #2: Yes

5. Is the manuscript presented in an intelligible fashion and written in standard English?

Reviewer #1: Yes

Reviewer #2: Yes

6. Review Comments to the Author

Reviewer #1: The authors have fixed the issues from the first review, and the paper is much clearer now. The idea that eco guilt works in a non linear way makes more sense than calling it a confounder. The methods are explained better, with a clear translation process and the addition of VIF, which makes the results more reliable. The discussion is also stronger, especially the idea that ecovanxiety can be helpful and the step by step path from grief to reflection to anxiety to guilt. The authors are careful with their claims and clearly mention the limits of using Dutch and German samples. Overall, the paper is well written and gives useful insights into why people act in pro environmental ways. For future work, it would be good to test this model over time to better understand how these emotions develop.

Reviewer #2: Dear Author, thank you for your efforts in addressing my comment. I really appreciate your time and effort.

7. PLOS authors have the option to publish the peer review history of their article (what does this mean?). If published, this will include your full peer review and any attached files.

Reviewer #1: No

Reviewer #2: No

---

## [Editor Report · Acceptance letter]

PONE-D-25-59455R1

PLOS One

Dear Dr. Petkovski,

I'm pleased to inform you that your manuscript has been deemed suitable for publication in PLOS One. Congratulations! Your manuscript is now being handed over to our production team.

Kind regards,

on behalf of

Dr Zhengmao Li

Academic Editor

PLOS One